# Legends, Inspirations and Space: Landscape Sacralization of the Sacred Site Mount Putuo

Yiwei Pan [1,2] and Aibin Yan [1,*]

1  Department of Landscape Planning and Design, East China University of Science and Technology, Shanghai 200237, China; panyiwei@mail.ecust.edu.cn
2  Shanghai World Expo Land Holdings Co., Ltd., Shanghai 200125, China
*  Correspondence: yanaibin@ecust.edu.cn

**Abstract:** Mount Putuo in Zhejiang Province, China, is the most important holy land of Guanyin in East Asia. Landscape sacralization is a key modality by which sacred meaning is constructed. This paper takes several examples—the Tidal Sound Cave ("chaoyin dong" 潮音洞), the Well of the Immortal Mei ("Meixian jing" 梅仙井), the Well of Ge Hong ("Ge Hong jing" 葛洪井), the Well of the Immortal ("xianren jing" 仙人井), and Duangu Pier ("Duan Gu daotou" 短姑道頭)—to analyze the three types of processes of sacralization. The Tidal Sound Cave is a re-construction of the founding myths; Well of the Immortal Mei, the Well of Ge Hong and the Well of the Immortal reflect harmony between local legends of Daoist immortals and the sacred Buddhist site; and the Duangu Pier accomplished its sanctification process in the course of local pilgrimage activities. By sorting out the mechanism and process of landscape sanctification and exploring the generation and renewal of landscape meaning, we can observe the logic of the construction of this sacred site.

**Keywords:** sacred site; Mount Putuo; Guanyin; legends; inspirations; space





## 1. Introduction

Mount Putuo, an island in the east of Zhejiang Province, China, is an important pilgrimage site of Guanyin in the East Asian cultural circle and is one of the four famous Buddhist mountains ("sida mingshan" 四大名山) in China, along with Mount Wutai in Shanxi, Mount Emei in Sichuan, and Mount Jiuhua in Anhui. Historical literature dates the legends of Mount Putuo to the Tang Dynasty (618–907) and earlier, but it was in 1080 that the monastery on the island was officially recognized and called Baotuo Guanyin Temple 寶陀觀音寺. From 1080 to the Qing Dynasty (1636–1912), due to the island's special geographical location, monasteries on Mount Putuo have risen and fallen unpredictably, but three stages of changes to general spatial patterns can be seen: Baotuo Guanyin Temple as the single center (1080–1606); Putuo Temple 普陀禪寺 and Zhenhai Temple 鎮海禪寺 as the front and rear centers of the island (1606–1793); and Puji Temple 普濟禪寺, Fayu Temple 法雨禪寺 and Huiji Temple 慧濟禪寺 as the three major temples (from 1793) (see Ni 2018, pp. 128–36). Temples provide places for religious practice, and in parallel with the monastic changes, some landscapes participated in the construction of the sacredness of the holy site (Figure 1).

Over time, legendary stories contribute meaning to a site, as does experience of a site. Stories of sightings of Guanyin on Mount Putuo formed an important founding myth. Subsequently, inspirations have been continuously recorded and spread through gazetteers and other media and taken as solid evidence for the presence of Guanyin on the site. "Inspiration" (linggan 靈感) here means happenings attributed the divine power of Guanyin. Some miracle stories that occurred on various sites may replay in later generations with similar stories about different people, resulting in several centers of inspiration; other sites are considered to be associated with legends because of their naturally unique topographies or how space is experienced. The significance of both

natural and manmade landscapes develops with use, becoming sacralized. In this paper, this process is called as "landscape sacralization". How that happens is the topic of this paper.

Several scholars have analyzed the island and discussed its meaning. Yü Chün-fang introduces how Mount Putuo became the sacred site of Guanyin in early times, and she also discusses the founding myths and the miracle of sighting Guanyin (Yü 2001, pp. 383–88). Marcus Bingenheimer points out that the meaning of the sacred site exceeds the location; text and site form a feedback loop (Bingenheimer 2016, pp. 12, 13). He also analyzes some examples, such as the Tidal Sound Cave, the Brahma Voice Cave, and the Sudhana Cave. Ni Nongshui, on the other hand, analyzes the cultural meanings behind the inspiration stories of the site, arguing that the miracles reflect relationships between the holy island and the royal families, officials, and ordinary worshippers (Ni 2018, pp. 190, 191). All these discussions provide useful references for this paper.

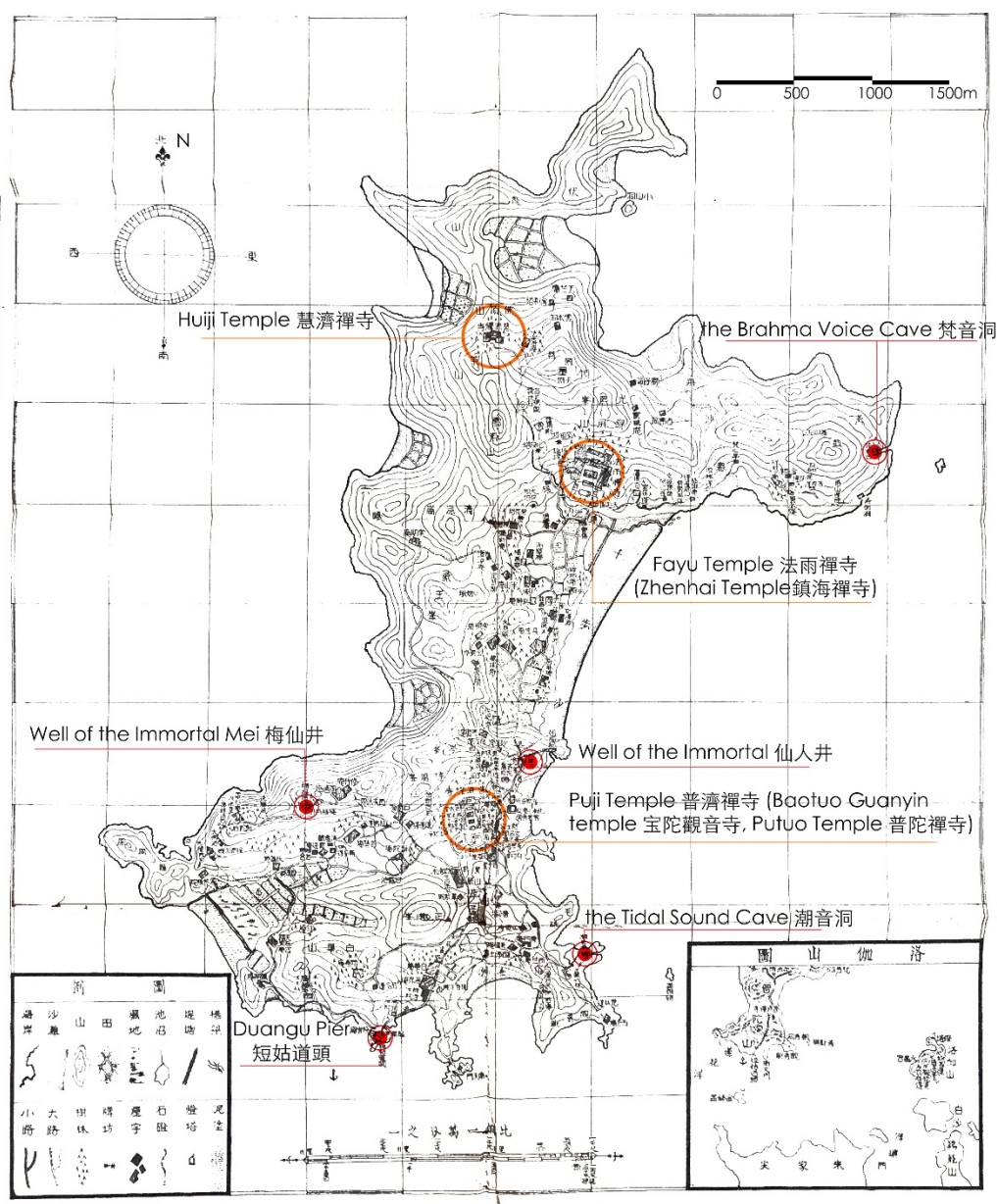

**Figure 1.** Plan of Mount Putuo, dating to between 1908 and 1949 ("Putuoshan quan tu" 普陀山全圖. "Shi yin ben" 石印本. Harvard Yenching Library Rare Book T 31008728.).

This paper discusses three modes of landscape sacralization. Natural landscape and manmade landscape realize sacralization and three kinds of sacred sites: holy natural site, landscape group or cultural center. Miracle stories are the key link in the process. Three typical cases are used in the following discussion corresponding to the three modes.

This study focuses on stories whose internal sequence of events is often not consistent, however. Miracle stories in texts are often accompanied by specific times or years, but no single story can be said to consistently follow a chronology. For example, iterations of the well-known story "Buddhist Establishment by Monk Egaku 慧鍔" ("Hui E Kai Shan" 慧鍔開山) are largely the same in detail but those details vary widely in chronology[1]. Therefore, this paper focuses on how legends were carried in specific spaces in gazetteers and other texts, how later inspirations echoed the earlier ones, and how they were developed and fleshed out in the context of the sacred sites, trying to get as close as possible to the content itself without examining chronology.

## 2. From "Foreign Monk Burning Fingers" and "Buddhist Establishment by Monk Egaku" to the Establishment of the Tidal Sound Cave's Inspiration Status

The Tidal Sound Cave (Figure 2) was the most important inspirational place on Mount Putuo prior to the Kangxi era (1662–1722). It played a key role in the construction of the sacred site of Mount Putuo. A story about the cave records what is regarded as the first miracle of Mount Putuo, about a foreign monk in the Dazhong 大中 period (847–860) of the Tang Dynasty. It appeared in the first gazetteer of Mount Putuo, *Butuoluojia shan zhuan* 補陀洛迦山傳 (1361). The monk in this story burned his ten fingers, one after another, in front of the Tidal Sound Cave, and then he saw Guanyin and was awarded precious stones (*Taishōshinshū daizōkyō* T 51, p. 1136c)[2]. "From then on," we read, "people who pray sincerely may get responses from Guanyin. She sometimes appears in purple and golden appearance, wearing a white dress with silk belt and multi-colored beads; sometimes she appears with a thousand heads and arms, guarded by lokapala [deities who protect Darma]." (Xu 2002, pp. 9b–10a) Although the legend "Foreign Monk Burning Fingers" ("Fan Seng Ran Zhi" 梵僧燃指) cannot be found in firsthand documents of the Tang Dynasty, by at least the Northern Song Dynasty (960–1127), after Buddhism was actually established on the island, the Tidal Sound Cave had become the first natural landscape to be sanctified. In another legend, "Buddhist Establishment by Monk Egaku," according to *Butuoluojia shan zhuan*, the Tidal Sound Cave appears again. The Japanese monk Egaku's boat struck a reef and could not move forward, so he knelt piously in the direction of the Tidal Sound Cave and safely reached the shore (T 51, p. 1136c). This record seems to echo the "Foreign Monk Burning Fingers" legend. The latter story implies that Egaku knew of the direct connection between the Tidal Sound Cave and Guanyin, which is why he chose to kowtow toward that natural site.

In addition to the inheritance relationship between the two stories above, we can also see two different modes of Guanyin's manifestation. In the first story, the foreign monk saw Guanyin and received precious stones; in the second story, the record does not mention whether Egaku saw Guanyin but simply that Guanyin helped him out of danger. Although the precious stones he received did not become part of the legendary system, nor did they develop as material remains or manmade spaces, Egaku enshrined the event by bestowing Guanyin status upon a private house on the side of the Tidal Sound Cave, whose homeowner later donated the house and built a monastery called "Guanyin Who Refuses to Leave" ("Bukenqu Gaunyin Yuan" 不肯去觀音院). This temple has not survived, though it is mentioned in historical texts, and in the 1980s a new temple was built in a similar location with the same name.

The sacralization of the Tidal Sound Cave landscape can be explored through the concept of a palimpsest. In landscape research, discussions about the layering of meaning have accumulated in the West. In the past, when paper was precious, an existing text might be erased to make way for a new one, often leaving traces, and the layering of different traces that could be detected in one manuscript is called a palimpsest. In recent research

about landscapes, the idea of treating a landscape as a palimpsest means identifying various physicalities or traces within it (Doherty 2016, p. 29). The influence and interpretation of the above-mentioned two modes of Guanyin's manifestation that are evident in the sacralization of the Tidal Sound Cave can be also seen in other inspiration stories, which together can be understood as one type of palimpsest. In addition, traces of multiple periods can be seen in the area around the Tidal Sound Cave, not in the form of relics but internalized within contemporary spaces. These historical traces correspond to multi-layered miracle tales.

Both characters, the foreign monk and the monk Egaku, are somewhat legendary figures. But in the Song Dynasty, miracles began to occur for a number of officials who were real historical figures. The following will focus on the experiences of Wang Shunfeng 王舜封 and Shi Hao 史浩 (1106–1194), to reveal the characteristics of inspiration stories and their role in spatialization. According to *Butuoluojia shan zhi*, during the Yuanfeng 元豐 period of the Song Dynasty (1078–1085), the emissary Wang Shunfeng prayed when he encountered wind and waves on his journey by water, and he "suddenly saw a golden shimmer, like a full moon beaming with a pearly luster, coming out from the rock cave, so he sailed smoothly (T 51, p. 1137a). This description is quite similar to the legend "Buddhist Establishment by Monk Egaku," but it does not indicate whether the rock cave is the Tidal Sound Cave. Later, in the Wanli 萬曆 period of the Ming Dynasty (1573–1620), the story recorded in *Chongxiu Putuoshan zhi* clearly states that Wang "passed the Tidal Sound Cave when the black wind suddenly rose" and he "kowtowed to the mountain and witnessed Guanyin" (Zhou 1980, p. 136). Not only is the location, the Tidal Sound Cave, specifically mentioned, but the manifestation of Guanyin is concretized as having "witnessed Guanyin," which is obviously mixed with the manifestation mode in the story "Foreign Monk Burning Fingers." Since then, inspiration stories about the Tidal Sound Cave share the similar experience of seeing Guanyin.

The story about Shi Hao says that he came to the Tidal Sound Cave on a morning in March of 1148. He saw auspicious signs in the cup when he served tea for the bodhisattva, but he did not see Guanyin. He came again in the afternoon but still did not see anything. Upon the guidance of a monk, he came to the hollow at the top of the rock cave, and "when he looked down at the cave, he suddenly saw the appearance of Guanyin, with clear features and shining in gold" (T 51, p. 1137a). This story reflects a difference from the above three stories. Shi Hao made a special trip to the cave to witness the manifestation of Guanyin, and his two visits in a single day imply that he wished to see Guanyin in person rather than being satisfied with auspicious signs. This legend initiated a new form of visiting the Tidal Sound Cave—from the top of the rock. The hollow is a called "Sky Window" ("Tianchuang" 天窗) (Figure 2), a new term that became attached to the cave and was specifically marked in gazetteers and also in the painting of Sacred Land of Mount Potalaka (*Butuoluo shan shengjing tu* 補陀落山聖境圖)[3].

The Tidal Sound Cave is located at the southeast corner of the island and is a cave formed by sea erosion. Sea water rushes into the cave during high tide, so it is quite dangerous to go into the cave, and looking down from the "Sky Window" is objectively safer. The area around the cave is full of uneven rocks, a difficult place to walk through. A bridge called "Dashi Bridge"[4] was built in the Southern Song Dynasty. In 1387, however, the government imposed a maritime embargo on Mount Putuo and destroyed almost all the temples on the island. Only one small shrine, with a roof of iron tile ("Tie Wa Dian" 鐵瓦殿), remained to continue the Buddhist tradition. This shrine was located near the Tidal Sound Cave (Tu and Hou 1589), indicating that the cave retained its high inspirational status during the Ming Dynasty.

The "Foreign Monk Burning Fingers" legend seems to indicate that one could see Guanyin by burning one's fingers. Buddhist scriptures, such as the *Wonderful Dharma Lotus Flower Sutra* (*Saddharma Pundarika Sutra*), also point out the merits of burning fingers, arms or bodies as offerings to the Buddha[5] (T 09, pp. 53c–55a). Although there is no record in gazetteers of later generations that worshipers followed the example of the foreign

monk and burned their fingers to obtain vision, the idea of sacrificing one's body becomes prohibited in later times, suggesting that this legend led to mimicking. We learn, in an article called "Exhortation of Prohibition against Sacrificing Bodies" ("She Shen Jie" 捨身戒), written by Dong Yongsui 董永燧, a commanding general in the Ming Dynasty, that the general built a pavilion named "Pavilion of Forbidden Living Sacrifice" ("Mo She Shen Ting" 莫捨身亭) near the cave. A memorial stele called "Forbidden Giving Bodies or Burning Fingers" ("Jinzhi Sheshen Ranzhi Bei" 禁止捨身燃指碑) (Figure 2) was erected by the Ming Dynasty officials Li Fen 李分 and Chen Jiusi 陳九思 in front of the temple "Bukenqu Gaunyin Yuan", reflecting the mania of the time of burning fingers at the Tidal Sound Cave. The late Ming literati Zhang Dai 張岱 recorded the scene he saw in the hall of Putuo Temple (the largest temple at that time) on the evening before Guanyin's birthday, February 19 of the Chinese lunar calendar: "Many nuns burned their heads, arms, and fingers on this night, and some young laywomen believers also followed them" (Zhang 1935, p. 47). Although he did not agree, he provided records of burning bodies and fingers on Mount Putuo in the Ming Dynasty. The "Forbidden Giving Bodies or Burning Fingers" stele declared, in the form of a rigid decree, the following: "If there are any foolish people who dare to sacrifice bodies or burn fingers at the Tidal Sound Cave, the abbot monk must forbid them. Repeated offenders will be apprehended and prosecuted." In contrast, Dong Yongsui's "Exhortation of Prohibition against Sacrificing Bodies" exhorted readers in a didactic way: Instead of sacrificing one's body for blessings, one should sacrifice the body for righteousness and practice present filial piety, loyalty and fraternity (Wang 1980, pp. 455–56).

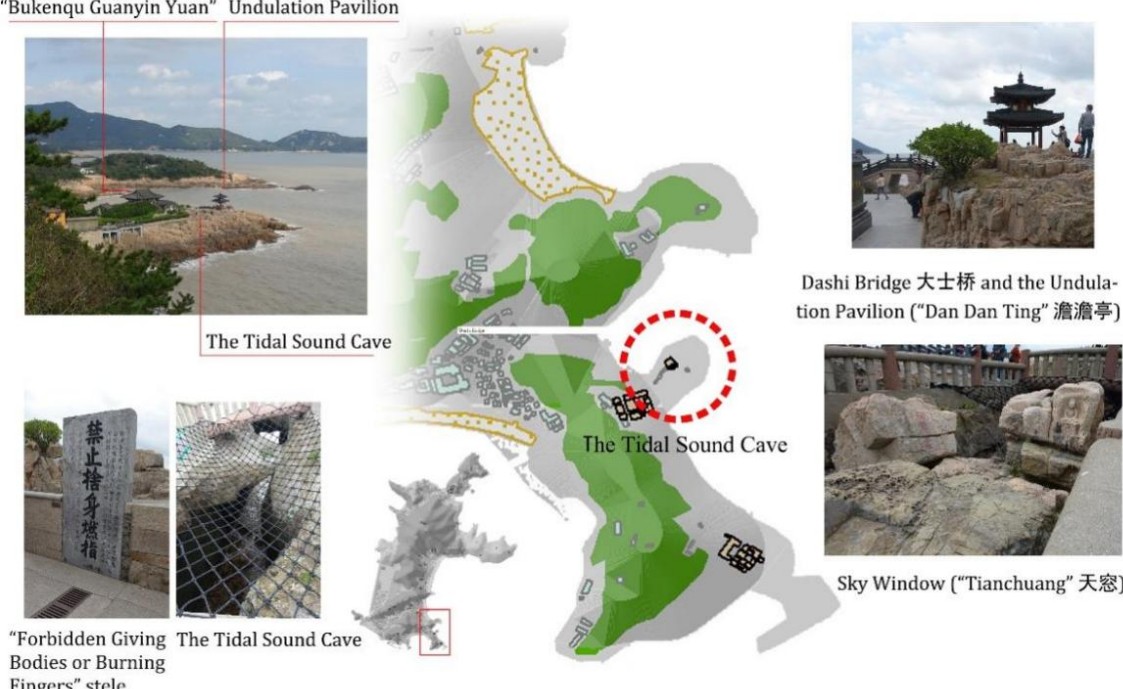

**Figure 2.** The location of the pier of the Tidal Sound Cave and the surrounding landscape.

The Tidal Sound Cave played a key role in the transformation of Mount Putuo from an ordinary natural island to first a site of inspiration and then to a sacred place of public pilgrimage. As the Tidal Sound Cave became established as a site of inspiration, the textual dissemination of inspiration stories interacted with activities of constructing space. The "Foreign Monk Burning Fingers" legend initiated a direct connection between Mount Putuo and Guanyin; the "Buddhist Establishment by Monk Egaku" legend laid the foundation for the construction of the sacred site. Due to the spread of holy stories, the images of Guanyin seen by previous generations had the potential to affect how later generations experienced

them as inspiring. In this process, the specific spaces evolved in three ways. First, the specific spaces from the founding myths of the mountain reappeared, mainly through the construction of "Bukenqu Gaunyin Yuan". Second, spaces developed at different times with specific significance to serve the status and function of Mount Putuo: the Dragon Palace, built at the entrance of the Tidal Sound Cave during the Song and Ming dynasties for rituals to pray for rain; the "Tie Wa Dian", built during the maritime embargo period of the Hongwu era; and the Undulation Pavilion ("Dan Dan Ting" 澹澹亭), built in 1980 for tourists to rest and view the sea. Third, some landscape elements were built to correct the misconceptions spread by inspiration stories, represented by the "Forbidden Giving Bodies or Burning Fingers" stele and the Forbidden Sacrifice Lives Pavilion. These three types of spatial sacralization contributed to the gradual transformation of the Tidal Sound Cave from a single natural cliff cave into a relatively complete landscape group.

### 3. Wells: "Material Evidence" of Local Legends of Daoist Immortals

The chapter "Buddhist Monks" ("shizi" 释子) in *Chongxiu Putuoshan zhi* (1607) lists the prestigious monks of Mount Putuo before the Wanli period of the Ming Dynasty (1368–1644), followed by four "virtuous Daoist masters": An Qisheng 安期生 in the Qin Dynasty, Mei Fu 梅福[6] in the Han Dynasty, Immortal Ge (Ge Hong) in the Jin Dynasty and Wang Tianzhu in the Yuan Dynasty. Later gazetteers also included the four masters in the legend system, and native miracles about them also participated in the process of spatialization of the sacred site.

The "Mei" of Mount Putuo's old name, "Meicen 梅岑," refers to Mei Fu, the first Daoist master who has been clearly recorded in gazetteers and the only Daoist master who practiced alchemy on Mount Putuo, as recorded in the first gazetteer *Butuoluojia shan zhuan*. The second half of the name, "cen 岑" means the small but high hill. The *Butuoluojia shan zhuan* (1361) says that "according to folklore here is the place where Mei Fu practiced alchemy" (T 51, p. 1136b). There are two scenic names related to Mei Fu in the painting of Sacred Land of Mount Potalaka, the Well of the Immortal Mei ("Meixian jing" 梅仙井) and the Alchemy Platform of the Immortal Mei ("Mei zhen liandan tai" 梅真炼丹台), indicating that, at least in the Yuan Dynasty, the legend of Mei Fu was incorporated into the special creation of the sacred site.

The sixth chapter of *Butuoluojia shan zhuan*, "Poems of Famous Sages," contains a poem by Liu Renben 劉仁本 (1308–1367), which says, "Mei Fu left the magical pellets as red as tangerines/An Qisheng gave jujubes as big as melons" (梅福留丹赤如橘, 安期送枣大于瓜). Here Mei Fu and An Qisheng are paired, but a direct connection between An Qisheng and Mount Putuo is not established. Both the two gazetteers of the Wanli period, *Butuoluojia shan zhi* (1589) and *Chongxiu Putuoshan zhi* (1607), copy in their entirety the first four chapters of *Butuoluojia shan zhuan* by Sheng Ximing, but *Chongxiu Putuoshan zhi* adds an additional passage, written by Liu Renben to the venerable master Seng Rui 僧睿 of Mount Putuo, which records that on October 6 of the Chinese lunar calendar in 1355, Liu visited Mount Putuo and saw White-Robed Guanyin in the Tidal Sound Cave and saw the general and arahants on the wall at the cave entrance. Liu compares Mount Putuo with the mountain on the sea where the immortals live, pursued by emperors of ancient times, saying that "near the place, there are the hometown of An Qisheng and capital of Penglai, and the scenery came clearly into view," thus proving that "what Sheng Ximing said is not lying." Liu points out that at his time, "the hometown of An Qisheng" was near Mount Putuo, an indication that the legend of An Qisheng had been distributed in Zhoushan at that time. According to Ni Nongshui's research, in the Song Dynasty literature, there was "An Qisheng Cave" at Mount Maqin 馬秦山 on Zhujia Jian 朱家尖 and Mount Maji 馬跡山 on Shengsi 嵊泗 Island (Ni 2018, p. 37). Both Wu Lai 吳萊 in the Yuan Dynasty and Tu Long 屠隆 in the Ming Dynasty said that Mei Fu directly practiced alchemy on Mount Putuo and that the legend of An Qisheng occurred in the area somewhere around the island. However, in *Chongxiu putuoshan zhi*, Zhou Yingbin lists An Qisheng as the first virtuous Daoist master of Mount Putuo, saying that he "came to the mountain to avoid

turmoil in the Qin Dynasty and sprinkled peach-blossom patterns while being tipsy, so there is a peach blossom mountain at the southwest of the temple" (Zhou 1980, p. 184).

While Mei Fu and An Qisheng are both representatives of the immortals, Ge Hong (283–343) is the representative of the theorists of longevity. Ge Hong is an important historical figure of Daoism, and his book, *Bao pu zi* 抱樸子, records the meditation and alchemy of the time. He proposes that the "outer alchemy" of minerals and the "inner alchemy" of nutritive essence, vitality and the spirit of the body itself may both help people achieve longevity. There are no legends about Ge in gazetteers of the Ming Dynasty, but Ge Hong's Well is spatial evidence of his story. The record about Wang Tianzhu in the Yuan Dynasty is even simpler; he "once practiced austerities in this mountain and later successfully prayed for rain; then he was known to the imperial court and given the assumed name 'Tai xu xuan jing zhi ren' 太虛玄靜志人" (Zhou 1980, p. 184).

The spread and development of immortal legends in Mount Putuo has dual significance, both as place metaphors and for historical construction. On the one hand, the unique natural landscape of Mount Putuo coincides with the place where the immortals of Chinese folklore lived in ancient times—the sacred mountains on the sea were believed to be sites for both immortals and elixirs, so the names of both Mount Meicen and Mount Putuo contain the word "mountain" instead of "island." On the other hand, although no circumstantial evidence connects the legends of Daoist immortals in Mount Putuo to it as a Buddhist sacred site, a connection with the local Daoist tradition had to be established in order to acquire inspiration that would appear inherent and legitimate and become widely recognized as a Buddhist site. Guanyin's residence represents the religious significance of the sacred site, but Buddhist literature is often unclear on how a site is related to the temporal dimension. In the historical imagination of ancient times, the famous mountains are often associated with Daoist legends, and to fully understand the meaning of the site, a local historical lineage perceivable by the Chinese would be useful. For these legends to play a lasting role in Buddhist sacred sites, they need both physical remains and coordination with Buddhism.

Zhou Yingbin used the terms "practices austerities" ("xiulian" 修煉) and "practices alchemy" ("liandan" 煉丹) to describe the behavior of four Daoist masters in Mount Putuo—An Qisheng and Wang Tianzhu "practiced austerities" whereas Mei Fu and Ge Hong "practiced alchemy." No names of scenic places or other spaces in Mount Putuo retain traces of An or Wang, whereas the scenic names associated with Mei and Ge can be seen as the Well of the Immortal Mei (from which later developed the Temple of Immortal Mei) and the Well of Ge Hong. These wells are "material evidence" of legends about Daoist immortals, and they both serve as symbols of the alchemy well ("danjing" 丹井) that echoes the legends and actually contains water that can be drawn. Wells related to Daoist immortals have further contributed their own inspiration stories. For example, the cave in which the Well of the Immortal is located is extremely chilly, like a Daoist retreat room (Figure 3). The water in the Well of the Immortal experiences "no increase or decrease in droughts or floods," and of the Well of the Immortal Mei, it "is said that washing eyes with the well water can make eyes clear" (Wang 1980, p. 124). During the Wanli period, the abbot monk of Puji Temple, Master Ji'an Rujiong 寂庵如迥, set up a hut called Meifu Temple beside the Well of the Immortal Mei, and in the early Qing Dynasty, the literati Lu Bao 陸寶 changed the name to the Temple of the Immortal Mei ("Meixian an" 梅仙庵), to avoid using the sage's full name (Figure 4). The pattern of establishing a related temple near a natural legendary site is quite similar to the formation of a temple around the Tidal Sound Cave.

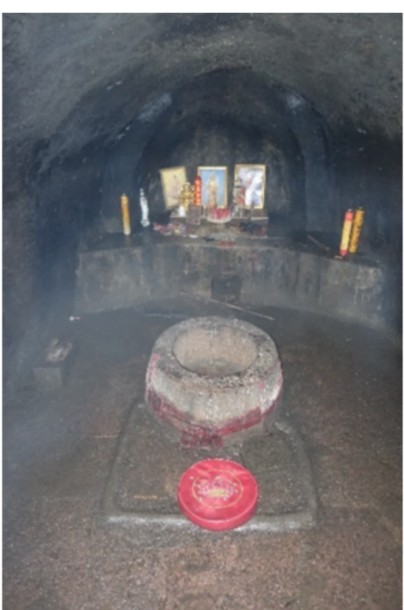

**Figure 3.** Well of the Immortal.

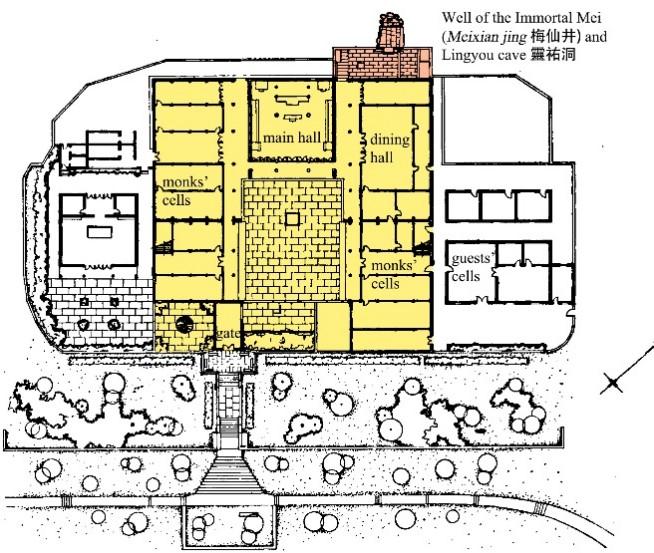

**Figure 4.** Plan of the Well of the Immortal Mei and the Temple of the Immortal Mei (Zhao and Ding 1997, p. 275).

The stories about Guanyin and the legends of the Daoist immortals represent two contexts. As a sacred Buddhist site, the legend from Daoism must contain some explanation that can be reconciled with Buddhist legends, otherwise it will easily fall into the deviation of attributing the sacred Buddhist site to Daoist origins. Master Yinguang 印光 (1862–1940) wrote a passage called "Stele Record on the Merit of Repairing the Well of the Immortal at Mount Putuo" ("qi Putuoshan Xianrenjing gongde bei ji" 砌普陀山仙人井功德碑記) in 1903 on behalf of Master Jieru 戒如 of Hongfa Temple 洪筏禪院, which began with the statement: "Guanyin bodhisattva always lives on this mountain; the Daoist immortals frequently reside here. Although in the early days the sculptures and Buddhist teaching had not arrived and ordinary people couldn't see the bodhisattva, the response body [Nirmanatkaya] had already resided here and Daoist immortals could always see the Buddha's light." Master Yinguang believes that the ancient monuments about Daoist immortals are spread according to the Buddhist teachings, and the well water is actually

the water of great compassion flowing from the bodhisattva's heart, and thus "is known for its efficaciousness" (Yinguang 1938, 4. pp. 14–15).

**4. Duangu Pier: From "Divine Trace" to "Renowned Place"**

In Wu Hung's work on Chinese ruins, he discusses four kinds of traces in the landscape: divine trace, historical trace, remnant site and renowned place, all of which are marks of the past and can coexist and shift from one to the other (Wu 2012, p. 64). Duangu Pier is a typical landscape that shifts from a divine trace to a historical trace and then to a renowned place, which is both the starting point of the formal journey and a place with multiple layers of imagination.

*Yun lu man chao*雲麓漫鈔[7] by Zhao Yanwei 趙彥衛 in the Southern Song Dynasty and the painting of Sacred Land of Mount Potalaka both referred to the landing place as "Koryo Pier," whereas in gazetteers of the Ming Dynasty the name became "Duangu Pier." According to Wang Liansheng's survey, the locations of the two piers are not the same (Wang 2003). When the name "Duangu Pier" appeared, the name "Koryo Pier" quickly faded from the literature. Later, as the structures and inscriptions around Duangu Pier became more abundant, it formed a renowned landscape space with special significance and served as the beginning of the narrative.

We cannot know from the literature why "Koryo Pier" was replaced with "Duangu Pier", but we can find Koryo saram's activities on the island in early times. *Mo zhuang man lu* 墨莊漫錄 by Zhang Bangji 張邦基 was written at the turn of the Northern and Southern Song dynasties, about half a century after Baotuo Guanyin Temple had been given plaque by the government. An article in the book records that the brass bells in the temple on Mount Putuo was given by Koryo merchants, on which engraved with Koryo's reign title. But records about Koryo saram's activities could not be seen in later literature. With the improvement of island's reputation, the government strengthened its control. Perhaps the change of the pier's name was related to a deep political significance. The name of Duangu Pier is related to an inspiration story about a pilgrimage of a pair of sisters-in-law, which was recorded in one version in *Chongxiu Putuoshan zhi* of the Ming Dynasty and in another in *Chongxiu Nanhai Putuoshan zhi* of the Qing Dynasty. According to *Chongxiu Putuoshan zhi*, there were two sisters-in-law who came to the mountain on pilgrimage. When the boat was about to land, the younger sister-in-law happened to have her menstrual period, so the elder sister-in-law went ashore and visited the mountain alone, leaving the younger one in the boat. After a while, an old woman came to the boat and used her lower garment to cover the stone walkway to guide the younger sister-in-law to the temple hall. When the elder sister-in-law went down the mountain, she found that her sister-in-law was no longer in the boat. After the younger sister-in-law came down the mountain, she told her sister-in-law about the experience. They went back to the temple hall to look for the woman but with no result. Then they realized that the old woman was a manifestation of Guanyin (Zhou 1980, p. 152). The story in *Chongxiu Nanhai Putuoshan zhi*, meanwhile, says that the two sisters-in-law came to Mount Putuo after several years of vegetarian diets, but when the younger sister-in-law had her menstrual period, the elder sister-in-law blamed her, and the younger sister-in-law was too ashamed to go ashore. As the tide rose, the stone path at the pier was flooded, and the younger sister-in-law on the boat was hungry and had no food to eat. At that time, an old woman carrying a bamboo container with food, threw a few stones into the water, walked to the boat, handed food to the younger sister-in-law and left. The younger sister-in-law was surprised and wondered who the old woman was. After a long time, the elder sister-in-law returned and was told the story. She guessed that it was Guanyin who appeared, so she immediately went back to the temple hall to pray and was surprised to see that the bottom of the lower garment on Guanyin's statue was wet. Since then, the pier has been called "Duangu Pier" (Pier of the sister-in-law who was blamed) (Xu 2002, pp. 5. 5a–5b).

The initial formation pattern of the Duangu Pier site is somewhat different from that of the Tidal Sound Cave. First, before it was named, the site was an artificially constructed

pier, on the basis of which inspiration stories were created and the pier was given a name. Second, the two main characters of the story, the pair of sisters-in-law, were ordinary female pilgrims who came to the island on their own, without the company of males. This kind of inspiration story of ordinary people is not common in gazetteers, and it is even rarer to develop a fixed name for a scenic spot based on the story of "nameless" female protagonists. Third, new inspiration stories were not repeated or attached to Duangu Pier in later times, and instead, the meaning of the space was enriched by means of inscriptions. The two versions of the story both reflect anxiety about female pilgrimages, and with the help of the female Guanyin, they provide two responses. In the story in *Chongxiu Putuoshan zhi*, the younger sister-in-law still successfully completed the pilgrimage under the guidance of Guanyin, which meant that women were liberated from inherent social prejudices and religious taboos with the help of the female bodhisattva. The story in *Chongxiu nanhai Putuoshan zhi* adopts a more conciliatory stance. The younger sister-in-law failed to complete the pilgrimage, but she nevertheless saw the manifestation of Guanyin and benefited from her presence. Regardless of the versions, the story reflects how Guanyin and Mount Putuo extend kindness to the common people, especially to female worshippers.

According to the story in *Chongxiu Putuoshan zhi*, Duangu Pier is "said to have been built by the bodhisattva herself, and has never been damaged by the pounding of huge waves" (Zhou 1980, p. 110). Zhou regards Duangu Pier as a divine trace left by Guanyin at the sacred site, which has non-historical characteristics. These characteristics can be seen in the woodblock printing "Sacred Relic of Duangu" ("Duangu shengji" 短姑聖蹟) from the series "Twelve Views of Mount Putuo" that appeared in the Qing Dynasty, in which Guanyin appears with a bamboo basket on the shore, rather than on the actual pier protruding from the coastline (Figure 5). The image reflected the imagination of the legend of a Guanyin manifestation rather than conveying the actual scene of the pier.

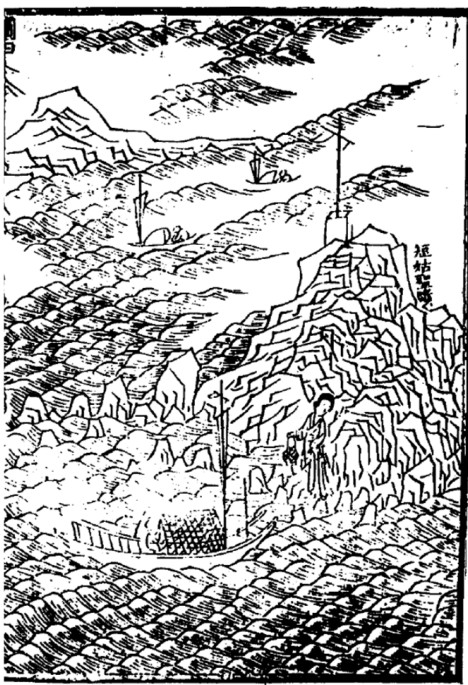

**Figure 5.** "Sacred Relic of Duangu" in the "Twelve Views of Mount Putuo" (Qiu 1996, p. 史 239–10).

There is no way to know when the story occurred, but the record in gazetteers and the inscription "Duan Gu gu ji 短姑古跡" (historical trace of Duangu) make this sacred site a historical trace. Inspiration stories recorded in gazetteers of the Qing Dynasty and the Republic of China were often arranged into a special chapter and in chronological order. The corresponding timeframe was pointed out at the beginning of each story. But not only does the story about the sisters-in-law in those gazetteers not include the exact time or

the dynasty in which it occurred, it also begins with the phrase, "according to legend." In *Chongxiu nanhai putuoshan zhi* by Xu Yan, the story is placed between the story from 1355 about Liu Renben and the story from 1403 about the manifestation of White-robed Guanyin in the Tidal Sound Cave. Evidently, the time of this story was considered to be the late Yuan to early Ming Dynasty. The stone inscription is another type of textual material. On the west side of the present Duangu Pier, standing in the sea, is a large stone engraved with the words, "Duan Gu gu ji", and the inscription "Shaohai bing Zhang Keda Dinghai dusi Gao Mingqian 紹海兵張可大定海都司高鳴謙" (Zhang Keda, the garrison commander of Shaohai and Gao Mingqian, the army officer of Dinghai) on the right and "Minguo shi'er nian daxun shi chongxiu 民國十二年大汛時重修" (rebuilt after the flood in 1923, the 12th year of the Republic of China) on the left. Both Zhang Keda and Gao Mingqian were from the late Ming Dynasty. The inscription seems to provide evidence that the words "Duan Gu gu ji" were recognized in the late Ming Dynasty. However, this inscription is not mentioned in any of the Ming and Qing Dynasty gazetteers, and the other stone inscriptions around it are all from the Republic of China (Figures 6 and 7). Although the records of the legend and the stone inscriptions are ambiguous in terms of the exact date, the legend of the sisters-in-law is objectively included in the historical narrative.

It is worth noting that the Chinese character "ji" in "Duan Gu gu ji" is "跡" with the "foot" radical (⻊), whereas the traditional characters for "跡" and "迹" are both unified as "迹" in simplified Chinese. Wu Hung points out the difference between the two characters. "跡" refers to a mark on the land whereas "迹" emphasizes movement, which means to leave one's own footprints when searching for traces of the past (Wu 2012, p. 63). The former character "跡" is used in "Duan Gu gu ji," whereas the latter "迹" is used in the writing of "spiritual traces" ("lingji" 靈迹) of the Tidal Sound Cave in gazetteers.

According to the record of Huang Yingxiong 黃應熊, who came to Mount Putuo in 1730, a wooden ornamental column ("huabiao" 華表) stood on Dangu Pier with the inscription "hai tian er fan 海天二梵" (Buddhist worlds of the sea and heaven). According to Wang Hengyan's record, this ornamental column was later rebuilt and the inscription was changed to "ci hang pu du 慈航普渡" (the barge of mercy ferries all the miserable people to the world of bliss) on the front and "fu hai wu ya 福海無涯" (the boundless sea of fortune) on the back (Wang 1980, p. 471). In addition to the structures and inscriptions, there are nearly ten inscriptions by people of the Republic of China around the stone inscription "Duan Gu gu ji." In addition, many publications about Mount Putuo from the Republic of China to the present have used the photograph of Duangu Pier as their cover. Although Duangu Pier has become a historical trace, it has also gradually become as renowned as a genius loci by regional consensus. As Wu Hung pointed out, the renowned place "cancels the historical specificity of individual traces" and became a place attracting visitors to leave their marks (Wu 2012, p. 86).

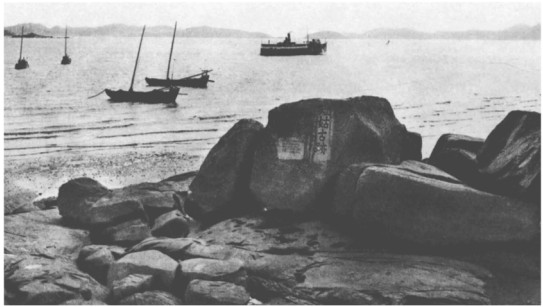

**Figure 6.** Stone inscriptions at Duangu Pier in 1930 (Yinguang and Zhenda 1930, p. 5).

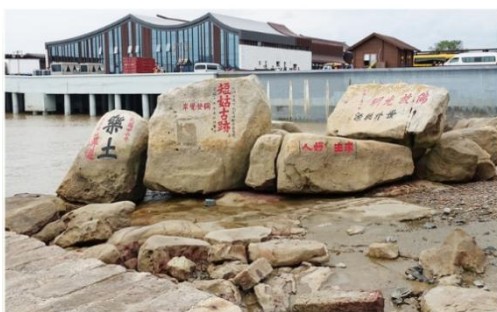

**Figure 7.** Stone inscriptions at Duangu Pier now.

## 5. Conclusions

Many legends of Mount Putuo show a reciprocal relationship with spaces of that landscape. The three modes of spatial interpretation discussed in this paper are particularly typical. The first mode is superimposing similar inspiration stories upon natural landscape and giving the natural landscape a special sacred meaning. In the course of the Tidal Sound Cave becoming the sacred center of the entire island, we can see how legends "Foreign Monk Burning Fingers" and "Buddhist Establishment by Monk Egaku" giving sacred meanings to the natural landscape. The second mode is forming a landscape space based on local Daoist legends, which in turn attaches new inspiration stories to the space, to the point where it gradually expands into a monastery. Wells named after Taoist immortals are the most representative. They both enrich the imagination of the history of the site and connect to Buddhist miracle stories. The third mode is inspiration stories based on the existing functional spaces and developing names for attractions and series of views. Duangu Pier is an excellent example. Legendary stories give the name and a unique religious significance to this island's indispensable functional space. In addition to the cases discussed above, other scenic spots on Mount Putuo can be classified according to these three modes.

The process of landscape sacralization requires a long passage of time. The sacred meanings of the landscape are transmitted and renewed through the use of space and the constant reenactment of inspiration. Sacred legends from ancient times to modern times, as listed by Yü in her important work (Yü 2001), are the soul of a sacred landscape. They maintain Mount Putuo as a site with sacred significance beyond its physical space, so that for centuries, throughout the East Asian cultural circle, the island has been recognized as the equivalent to Guanyin's dwelling of Mount Potalaka that is mentioned in the Buddhist sutra.

**Author Contributions:** Conceptualization, Y.P. and A.Y.; methodology, Y.P. and A.Y.; software, Y.P.; validation, Y.P. and A.Y.; formal analysis, Y.P. and A.Y.; investigation, Y.P.; resources, Y.P. and A.Y.; writing—original draft preparation, Y.P.; writing—review and editing, A.Y. and Y.P.; visualization, Y.P.; supervision, A.Y.; project administration, A.Y.; funding acquisition, A.Y. All authors have read and agreed to the published version of the manuscript.

**Funding:** This research was funded by [Shanghai Pujiang Program] grant number [2020PJC021].

**Conflicts of Interest:** The authors declare no conflict of interest.

## Notes

1     *Fo zu tong ji* 佛祖统纪 of the 13th century records the story "Buddhist Establishment by Monk Egaku" took place in 858, while *Butuoluojia shan zhuan* 補陀洛迦山傳 (1361) records the story took place in 916. *Shi shi ji gu lüe* 釋氏稽古略 of the 14th century followed the records of *Butuoluojia shan zhuan*.

2     T. *Taishōshinshū daizōkyō* 大正新修大藏經. Edited by Takakusu Junjirō 高楠順次郎 and Watanabe Kaigyoku 渡边海旭. 100 vol. Tokyo: Taishō issaikyō kankōkai, 1924–1935. *Butuoluojia Shan Zhuan* 補陀洛迦山傳. T 51, no. 2101. Edited by Sheng Ximing 盛熙明.

3   The painting Sacred Land of Mount Potalaka is preserved at the Jōshō Temple 定勝寺 in Nagano, Japan. Marcus Bingenheimer inferred that it was created between 1334 and 1369, the same period as mentioned by Sheng Ximing in *Butuoluojia shan zhuan* (Bingenheimer 2016, p. 57).

4   Dashi 大士 is a generic term used to call bodhisattva.

5   See "The Former Deeds of Medicine King Bodhisattva" (Chapter 23) of *The Wonderful Dharma Lotus Flower Sutra* (*Miao Fa Lian Hua Jing* 妙法蓮華經).T 09, no. 262. Translated by Kumārajīva.

6   Mei Fu 梅福 (*zi*, Zizhen 子真) was an education officer (*junwenxue* 郡文學) and Nanchang military commander (*wei* 尉). He repeatedly wrote directly to the emperor but was never accepted. In the period of Wang Mang 王莽, he abandoned his wife and children and went to Jiujiang 九江. Since then, he has been regarded as an immortal. Daoism regards Mei Fu as a member of its system of immortals.

7   *Yun lu man chao* 雲麓漫鈔 dates back to 1206, according to the inscription in the preface. This book records the names of landscapes in Mount Putuo from the Song Dynasty in some detail, so it is an important historical source for understanding the landscape during the Song Dynasty.

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
