# Peer review of "Legends, Inspirations and Space: Landscape Sacralization of the Sacred Site Mount Putuo"

_religions, doi:10.3390/rel12121050_

Round 1
Reviewer 1 Report
It is a poor start for any review to have to point out a typo in the title.
Though that might reflect more on the copy-editor than the author "scared" instead of "sacred" is definitely inauspicious. The maps and illustrations are all nicely done and very helpful to orient the reader.
- “sida mingshan” needs Chinese chars
- l37 "Some spiritual stories that occurred on various sites" wrong preposition apart, "spiritual stories", really?
- l49 in-spiration??
- l52 " pro-vide useful references for this paper.. " sic. I will in the following not comment on any more typos, wrong hyphenation, or extra punctuation.
- "The Tidal Sound Cave (Figure 2) was the most important inspirational place on Mount Putuo prior to the Kangxi era (1662-1722)" not after Kangxi?
- "In addition to the inheritance relationship between the two stories above" what is an "inheritance relationship"?
- “hai tian er fan 海天二梵.” why not translate this and the following two inscriptions?
The article seems unfinished. I do not see a research question. Neither does author present new information about the three sites, nor does he/she offer a convincing new interpretation or analysis. The presentation of facts for each place is unorganized and no attempt is made at a overarching narrative to tie the information together. Why these three places on Putuo and not some other sites on the island?
The "three modes" of "spacial interpretation" (l 432) appear too late in the conclusions, if this is the main point the paper should lead with them and justify the choice. Are the "spacial interpretation" the same as the "three types of spatial sacralization" (l 201)? Different modes of how legends are connected to places might be worth a paper, but the typology must be clearly thought through and discussed: Are there more modes than three? Is the first mode really one single mode or are their sub-modes? How do the other the sites on Putuo fit into this schema?
I would recommend to start again from scratch. 1) Find a convincing research question (or several) or interpretative paradigm and discuss it. 2) Then give the examples. Explain things and sources to readers. Phrases like "For example, iterations of the well-known story “Buddhist Establishment by Monk Egaku 慧鍔” (“Hui E Kai "))" (l.60) are not helpful to a wider readership. (There is a large literature about Egaku by the way that could be studied or at least cited). 3) Try to follow general English usage: "inspiration stories" (感應故事?) e.g. does not work in English, there are much such cases.
Author Response
Dear reviewer:
Thank you for your comments concerning our manuscript “Legends, Inspirations and Space: Landscape Sacralization of the Sacred Site Mount Putuo”. These comments are all valuable and very helpful for revising and improving our article, as well as the important guiding significance to our research. We have read through comments carefully and have made some revision. The main revision and responds are as following:
- Thank you very much to point out the diabolical errors. We have revised the typo in the title.
- We have add Chinese chars“四大名山” for “sida mingshan” .
- We agree that the phrase “spiritual stories” and “inspiration”may lead to misunderstandings in English. You have make the right translation “感應故事”. In order to avoid puzzles, we change the phrase to “miracle stories”.
- We have deleted unnecessary hyphen dashes in the paper.
- It’s true that the Tidal Sound Cave was the most important inspirational place on Mount Putuo prior to the Kangxi era (1662-1722) not after Kangxi era. This is because the Brahma Voice Cave (“chaoyin dong” 潮音洞)rose from1690s. After that, there are two inspirational places on the island. The course the Brahma Voice Cave
- About the “inheritance relationship”. Actually, it refers to the last sentence of the previous paragraph. Monk Egaku kowtowed toward the Tidal Sound Cave implied that he had known that the site was related with sacred meanings. According to earlier records we can see today, “Foreign Monk Burning Fingers” is the only story related to the cave.
- We have translated the inscriptions in this version.
- You raised a quite good point to the paper and to the whole research. Especially, we would express our gratitude to your comprehensive and profound amendment ideas. In this version, we expand the introduction to foreshadow our ideas more clearly. We tried our best to improve the manuscript and made some changes in the manuscript. But we remain the overall framework of the paper after careful consideration. Some point you raise have already been considered in our whole research about the landscape and architecture history of Mount Putuo,but this article mainly intends to show how landscape obtain sacred meanings. We don’t think that this paper can offer all-embracing things about Mount Putuo, so the three cases we choose in the draft intends to offer perspectives to the topic of sacralization. Even so, your valuable comments are still very helpful to our further research.
Once again, we earnestly appreciate for your warm and wonderful work, and hope that the correction will meet with approval.
Reviewer 2 Report
This is a solid piece of research grounded in textual analysis of primary sources. I recommend the publication of this paper after minor revisions. But, I found that this paper does not fit into the special issue, "Buddhist Architecture in East Asia." It seldom discusses the built environment of Mount Putuo and does not engage with the Buddhist architecture of the island per se. Instead, it delves into the role of legends and inspiration in the transformation of Mount Putuo into a sacred abode of Guanyin. I believe that it is better to move this paper to a regular issue of the Religions rather than having it published in this special issue. If the author wants to publish it in this particular issue, the author should examine the architectural aspects of the specific sites under discussion as well as actively engage in previous studies in the field of architecture and art history. For example, the author may discuss what role the legends under discussion played in the formation of material remains or manmade spaces on the island.
There are some terms and issues that should be clarified:
- Throughout the paper, the author uses the term "landscape sacralization." Since it is central to the author's main argument, it should be unpacked from the beginning of the paper outright.
- Line 187, the author uses the term "inspiration stories" but it is unclear what kind of literary genre that it corresponds to. I am not sure whether this is a standard term for the literary genre. Therefore, it is hard to follow the author's discussion. It would be helpful that the author provides Chinese characters for the term and explains it in the text.
- Some words have an unnecessary beginning. Eg.) Line 38 "in-spirations"; line 41 "de-velops"; one 65 "con-tent", etc.
- Line 217: English translation and Chinese characters for "Butuoluo shan shenjing tu" should be given.
- Line 234: "capital" appears to be a miswrite of "capital."
- Line 320: The author mentions that "Koryo Pier" was replaced with "Duangu Pier" but he/she did not discuss why this change happened or what this change meant in the larger historical trajectory of the island. Given the author's emphasis on the role of legends and inspiration stories, this particular case should be given more pages.
Author Response
Dear reviewer:
Thank you for your comments concerning our manuscript “Legends, Inspirations and Space: Landscape Sacralization of the Sacred Site Mount Putuo”. These comments are all valuable and very helpful for revising and improving our article, as well as the important guiding significance to our research. We have read through comments carefully and have made some revision. The main revision and responds are as following:
- Thank you very much for you appreciation. We consider that architecture is a part of landscape in a broad sense. Landscape plays a role as a significant context for architecture, and thus it is an integral part of architectural knowledge. This articlefocuses on how Buddhist relics obtain meanings, which may provide a framework for understanding single building. Coincidentally, we once wrote to the editors to consult whether our article is suitable for this special issue several months before, and got a positive reply. Sincerely hope that we can get your understanding and support.
- The second paragraph of the introduction explain the phrase “landscape sacralization”. In this version, we strengthen the term in the end of this paragraph.
- We have explained the word “inspiration” in the first paragraph of this version.
- We have deleted unnecessary hyphen dashes in the paper.
- We have added English translation for “Butuoluo shan shengjing tu補陀落山聖境圖”.
- We have revised the spelling mistake of “capital”.
- We have added a paragraph to illustrate the relationship and ratiocination of “Koryo Pier” and “Duangu Pier”.
Once again, we earnestly appreciate for your warm and wonderful work, and hope that the correction will meet with approval.